# Solid State Additive Manufacturing of Thermoset Composites

**DOI:** 10.3390/polym16172416

**Published:** 2024-08-26

**Authors:** Bo Hong, Kaifeng Wang, Yang Li, Shuhan Ren, Peihua Gu

**Affiliations:** 1Key Laboratory of Mechanism Theory and Equipment Design of Ministry of Education, Tianjin University, Tianjin 300354, China; bhong@stu.edu.cn (B.H.);; 2Intelligent Manufacturing Key Laboratory of Ministry of Education, Shantou University, Shantou 515063, China; 3International Institute for Innovative Design and Intelligent Manufacturing Tianjin University in Zhejiang, Shaoxing 312000, China; 4School of Materials Science and Engineering, Tianjin University, Tianjin 300354, China

**Keywords:** cold spray, additive manufacturing, thermosets

## Abstract

Softening and subsequent deformation are significant challenges in additive manufacturing of thermal-curable thermosets. This study proposes an approach to address these issues, involving the preparation of thermosetting composite powders with distinct curing temperatures, the utilization of cold spray additive manufacturing (CSAM) for sample fabrication, and the implementation of stepwise curing for each component. To validate the feasibility of this approach, two single-component thermosetting powders P1 and P2 and their composite powder C were subjected to CSAM and stepwise curing. From the sample morphology observation and deposition/curing mechanism investigation based on thermomechanical analysis and differential scanning calorimetry, it is found that severe plastic deformation occurs during the CSAM process, accompanied by heat generation, leading to local melting to promote a good bond at the contact surface of the particles and form small pores. During the progressive curing, the samples printed using C demonstrate superior deformation resistance compared with those using P1 and P2, and the curing time is reduced from 16.7 h to 1.5 h, due to the sequential curing reactions of P1 and P2 components in composite C, allowing the uncured P2 and cured P1 to alternately remain solid for providing structural support and minimizing deformation.

## 1. Introduction

Thermosetting materials have garnered significant attention in various industries, including aerospace, automotive, and electronics, due to their exceptional mechanical properties and chemical stability [1,2,3]. In recent years, there has been increasing interest in additive manufacturing of thermosetting materials, with photocurable resins at the forefront of this research trend [4,5]. However, the exploration of thermal-curable thermosets in additive manufacturing remains relatively underexplored.

One of the primary challenges in additive manufacturing of thermal-curable thermosets is their tendency to soften and deform [6]. This problem can arise both during the 3D-printing process and the subsequent heat-curing stages. Some traditional additive manufacturing methods rely on the deposition of liquid materials, such as Material Jetting (MJ) [7,8], Direct Ink Writing (DIW) [9,10], and Liquid Deposition Modeling (LDM) [11]. The inherent mechanical weakness of liquid materials makes 3D-printed structures easily deform under their own weight or external forces. This problem is particularly pronounced in materials with slow curing rates, as prolonged curing time increases the likelihood of deformation.

In addition, conventional additive manufacturing techniques using solid materials, including Fused Deposition Modeling (FDM) [12,13], Selective Laser Sintering (SLS) [14,15], and Fused Granular Fabrication (FGF) [16,17], typically involve rapid melting, liquid-state deposition, and rapid solidification of materials. In cases where complete curing is not achieved during the rapid solidification phase, subsequent heat treatment should be performed. However, this additional heating process can lead to further melting and deformation of the printed structures.

To address these challenges, an innovative three-step approach is proposed in this paper. Firstly, a composite powder material by combining multiple thermosetting materials with distinct curing temperatures is prepared. Secondly, CSAM, a technique that utilizes high-velocity particle impact for material deposition [18,19], is adopted, which can maintain the deposited material in a solid state throughout the 3D printing process [20,21]. Finally, a stepwise isothermal curing process is implemented, allowing the different components of the composite powder to melt and cure sequentially at different temperatures. This strategy ensures that only partial melting occurs in the overall material, thereby controlling deformation.

To validate the efficacy of this approach, a case study is conducted using two thermosetting powders, P1 and P2, and their composite powder C. Cylindrical samples are fabricated using CSAM technology and subsequently subjected to stepwise curing. The results demonstrate that the samples maintain their original shape throughout both the additive manufacturing process and the heat curing stages. Notably, the composite material C not only preserves its shape stability but also achieves complete curing in significantly less time compared to traditional low heating rate curing methods, thereby substantially enhancing production efficiency.

## 2. Materials and Methods

### 2.1. Materials

Two commercial bisphenol-A epoxy resin powders and their corresponding latent curing agents from Guangzhou Shinshi Metallurgical and Chemical Co., Ltd. (Guangzhou, China) were utilized to prepare thermosetting powders in this paper. The epoxy resins were designated as E50 and E57, while the curing agents were labeled as A80 and A95. Both the epoxy resin and the curing agents exhibit average particle sizes within the range of 3~5 μm. The epoxy equivalent weight (EEW) of E50 was 0.09~0.14 mol/100 g, whereas E57 had an EEW range of 0.05~0.06 mol/100 g. A80 and A95 were activated after melting and underwent curing reactions with epoxy resin.

The epoxy resins and curing agents were combined to create two distinct thermosetting powders, named P1 and P2. To ensure a significant difference in melting points between P1 and P2, thermomechanical analysis (TMA) was conducted on the resin and curing agent powders, as depicted in Figure 1. It can be seen that E50 and A80 exhibit similar melting ranges, as do E57 and A95. Consequently, E50 was paired with A80 to form P1, while E57 was paired with A95 to form P2. Both A80 and A95 were latent curing agents that activate upon melting, resulting in distinct activation temperatures for P1 and P2. According to the manufacturer’s technical data, the mass ratio of E50 to A80 in P1 was 4:1, and the same ratio was applied to E57 and A95 in P2.

### 2.2. Powder Fabrication Process

The powder preparation process was conducted in two distinct stages. Initially, single-component thermosetting powders P1 and P2 were fabricated using E50, E57, A80, and A95. Subsequently, composite powder C was fabricated by combining P1 and P2, as illustrated in Figure 2. The fabrication of P1 and P2 involved several steps: the epoxy resins and curing agents were mixed and then compressed, followed by pulverization in water and a drying process. The preparation of composite powder C followed a similar procedure, with P1 and P2 first mixed at a 1:1 volumetric ratio, then compressed, pulverized, and dried. All operations were conducted at room temperature to prevent curing reactions.

Several devices were utilized for the powder preparation, as presented in Figure 3. For the mixing process, a small drum-type powder mixer (KeLe, MIXER-2, Zhengzhou, China) equipped with stirring blades was employed, as shown in Figure 3a. When E50 was mixed with A80, E57 with A95, and P1 with P2, the mixer operated with a bi-directional rotation frequency of 0.5 Hz for a duration of 1 h. Compression was performed using a manual hydraulic press (YueJia, Press-15, Tianjin, China) with a cylindrical mold of 50 mm internal diameter, as shown in Figure 3b. The mixed powder was poured into the mold and subjected to a pressure of 30 MPa for 2 min, resulting in cylindrical bulks. As observed in the cross-section of the compacted powder in Figure 2, the powder particles were fully deformed and interlocked. Pulverization was carried out using a 1650 W blade grinder (FangKe, FS400, Changzhou, China) equipped with cross-shaped rotating blades, as shown in Figure 3c. The cylindrical bulks were first broken into pieces with a hammer, and then processed in the grinder along with water. The grinder was activated three times for 10 s each time, with each grinding interval of 1 min. Water was added to prevent curing reactions due to a rapid increase in temperature during grinding. After pulverization, water was first filtered out of the ground material with a filter cloth. The resulting wet powder was then placed in a forced air dryer (SuBai, 101-16, Shaoxing, China) at 30 °C for 24 h to ensure that the water was completely removed, as shown in Figure 3d. The average particle sizes of P1, P2, and C powders were between 45 and 50 μm, and Figure 2 depicts their irregular morphologies.

### 2.3. Three-Dimensional Printing Procedure

CSAM was conducted using a customized low-pressure cold spray system, as illustrated in Figure 4. The powders were controlled using a vibratory feeder and introduced into the nozzle, which has an exit diameter of 5 mm. The powders were then accelerated by air pressurized to 0.8 MPa and then deposited onto a nylon-66 sheet substrate.

Cylindrical samples for P1, P2, and C were fabricated during the CSAM process at room temperature. The nozzle was initially fixed at a specific point on the substrate for cold spraying. It is noticed that during the powder deposition process, the accumulation rate of the deposited material was approximately 2 mm/s when the air pressure was 0.8 MPa. To maintain a constant stand-off distance between the nozzle exit and the top surface of the samples being deposited (10 mm in this study), the nozzle movement was controlled vertically with the speed of 2 mm/s, facilitating the layer-by-layer construction of the cylindrical samples.

### 2.4. Curing Process

Thermal curing was performed on the 3D-printed samples, and two distinct curing methods were utilized to evaluate the impact of different curing processes on material deformation. The first method controlled the softening deformation by adjusting the heating rate. Based on the manufacturer’s data, a rapid heating rate of 2 °C/min and a slow heating rate of 0.1 °C/min were selected, with the heating temperature ranging from 50~150 °C. The impact of these heating rates on material deformation was then compared. The second method utilized a stepwise isothermal curing process to control the softening deformation by sequentially curing P1 and P2. To determine the appropriate heating temperatures, TMA was performed on P1, P2, and C, as depicted in Figure 5. Significant deformation was observed for P1 and P2 at temperatures exceeding 75 °C and 90 °C, respectively, indicating the onset of melting and curing. Consequently, the first isothermal heating temperature was set at 85 °C to ensure the curing of P1 while keeping P2 in a solid state. Once P1 was fully cured, the temperature was elevated to 150 °C to facilitate the rapid curing of P2.

To determine the heating duration for each stage, Vickers hardness measurements were conducted to assess the degree of curing, as shown in Figure 6. The Vickers hardness of P1 and P2 remained constant approximately after a specific heating time, indicating the completion of the curing reaction. Therefore, the process for determining the stepwise isothermal curing procedure is as follows: initially, heating was conducted at 85 °C for 1 h, followed by heating at 150 °C for 30 min. Prior to reaching the constant heating temperature, a ramp-up rate of approximately 20.1 °C/min was employed. The detailed heating schedule is illustrated in Figure 7.

### 2.5. Experimental Characterization

#### 2.5.1. Microstructural Analysis

The surface features and fracture surface morphologies of the thermosetting powders and cold-sprayed samples before and after heating were characterized using the Hitachi SU-1510 (Tokyo, Japan) scanning electron microscope (SEM). Notably, all the samples were fabricated using CSAM with powders P1, P2, or C, and the corresponding fracture surface was obtained by breaking the samples using pliers. After coating a thin layer of platinum on the sample surface to enhance conductivity, the SEM observation was conducted following the ISO 16700:2016 standard [22].

#### 2.5.2. Mechanical Property Analysis

To evaluate the changes in mechanical properties, the Vickers microhardness was assessed. The tests were performed using a microhardness tester (HuaYin, HV-1000B, Shaoxing, China) under an applied load of 10 gf for a dwell time of 15 s, in accordance with the ASTM E384-17 standard [23]. Three random regions without pores were tested for each sample to ensure the accuracy and reliability of the measurements. The hardness testing of powder materials was conducted prior to the crushing stage in the powder fabrication process.

#### 2.5.3. Thermo-Property Analysis

Differential scanning calorimetry (DSC) was employed to investigate the thermal properties and curing behavior of the printed samples, including P1, P2, and C, both before and after the stepwise heating conditions. The analysis was conducted using a DSC instrument (Shimadzu, DSC-60, Kyoto, Japan) under a nitrogen atmosphere with samples subjected to a temperature range of 25~200 °C at a heating rate of 10 °C/min.

In addition, TMA was conducted to investigate the material deformation at varying temperatures. The test samples were circular discs with a diameter of 20 mm and a thickness of 1~2 mm, fabricated from powder material and shaped using hydraulic pressing. The samples were placed on a programmable heated plate (BangQi, ET-100FBC, Shenzhen, China). During the test, a constant load of 20 g was applied using a spherical indenter with a diameter of 5 mm. For the softening range assessment, a heating rate of 10 °C/min was employed. When evaluating deformation under a stepwise isothermal curing process, the heating program depicted in Figure 7 was utilized. Throughout the TMA test, the indentation depth at different temperatures was recorded. The material deformation was characterized by the ratio of the indentation depth to the sample thickness.

## 3. Results and Discussion

### 3.1. Deposited Sample Morphology Analysis

Cylindrical samples were printed using CSAM technology with powders P1, P2, and C, as depicted in Figure 8a,e,i. These samples, with heights ranging from 6 mm to 12 mm and diameters of approximately 5.5 ± 0.5 mm, close to the nozzle diameter of 5 mm. The surface of the cylindrical samples exhibited a “wrinkled” structure, indicating instability during powder deposition and subsequently resulting in a change in deposition diameter. In addition, these parallel wrinkles suggest a consistent layer-by-layer deposition.

It is noteworthy that the deposition rates of powders P1, P2, and C in CSAM are 5%, 2.1%, and 3.5%, respectively, where the deposition rate is defined as the ratio of the mass of the powder material deposited on the substrate to the total mass of the powder spraying during the CSAM process. Due to the varying deposition rates of P1 and P2, spraying a mixture of these powders does not produce a deposited material in P1 and P2 proportions consistent with the original powder mixture. To mitigate the effects of these differences in deposition efficiency, this study utilized a compaction method to bond P1 and P2 relatively together to produce powder C.

Subsequently, the cylindrical samples were subjected to two distinct curing processes for curing. The first method was conducted by controlling the heating rate. As illustrated in Figure 8b,f,j, the samples printed using P1, P2, and C experienced significant softening deformation at a high heating rate of 2 °C/min. Conversely, when the heating rate was reduced to 0.1 °C/min, as illustrated in Figure 8c,g,k, the samples almost retained the cylindrical shape, although the heating duration extended to 16.7 h.

The second method utilized a stepwise isothermal curing process, as depicted in Figure 7, to sequentially cure the components and manage the deformation. Figure 8d,h illustrates that samples printed using P1 and P2 softened at their respective isothermal conditions of 85 °C and 150 °C. In contrast, the sample printed using C, as shown in Figure 8l, did not display any softening deformation with the stepwise curing process. It retained the cylindrical shape and the wrinkled surface texture observed in the initial morphology depicted in Figure 8i. Moreover, it is found that the corresponding stepwise curing process required only 1.5 h, significantly less than the time required by the low heating rate method (i.e., 16.7 h).

Therefore, employing composite powder material C for CSAM printing and combining it with a stepwise curing process can significantly mitigate the deformation issue both during the 3D printing and curing stages.

### 3.2. Deposition and Curing Mechanism Analysis

During the CSAM process, thermosetting powders P1, P2, and C can be successfully deposited at room temperature. At this temperature, P1, P2, and C are in a glassy state, exhibiting brittle characteristics. Traditionally, brittle materials have significant challenges for CSAM applications, primarily due to their tendency to fracture upon impact, which normally results in thin films rather than substantial deposits [24,25]. However, the CSAM samples printed using P1, P2, and C demonstrated the ability to form cylindrical structures, indicating their potential for additive manufacturing applications. Thus, the deposition mechanism of P1, P2, and C should be significantly different from that of other brittle materials.

To investigate the corresponding deposition mechanism of thermosetting powders during the CSAM process, the particle impact velocity was estimated first. The average particle velocity of P1, P2, and C powders at the nozzle exit was in the range of 378~387 m/s, as predicted by the one-dimensional steady gas-dynamic model [26,27]. This high velocity provided the necessary conditions for severe plastic deformation of the particles upon impact. The surface morphologies of the samples printed using P1, P2, and C were examined, as shown in Figure 9. It can be seen that all the samples printed using P1, P2, and C had a number of sharp ridges and valleys on the deposited surface, indicating that the heat generated during impact was insufficient to cause complete melting of the whole particle.

Meanwhile, the fracture surface of the samples printed using P1, P2, and C was observed, as shown in Figure 10a–c. It can be seen that the original form of the powder particles was no longer recognizable, and the contact interface between particles was also indistinguishable, indicating severe plastic deformation had occurred during the powder deposition. Additionally, a number of pores were found throughout the fracture surface of the samples, where the samples printing using P1 had the highest pore density, and the ones printing using P2 had the lowest pore density. Generally, the pores could be categorized into two types based on their shapes, i.e., irregular pores with diameters close to 20 μm and small round pores with diameters of a few micrometers. Similar to the cold spraying of metal powders [28,29], air could be trapped during the powder deposition process, leading to the formation of large irregular pores. For the dense small circular pores, they should be generated during the impact of the particles. Our prior research indicates that particles deposited onto the surface display lower melting peaks in DSC compared to the original powder particles [30]. This observation suggests that local melting occurred during impact, activating latent curing agents and accompanying release of blocking agents, where gas was generated during the decomposition or volatilization of the blocking agent to form the small circular pores [31]. Hence, the presence of these micro-sized pores implies that particle melting occurred at the impact surface.

Based on the above observations, the deposition mechanism of P1, P2, and C is due to the severe plastic deformation caused by high-speed particle impact, accompanied by significant heat generation. This thermal effect leads to local melting of the particle impact zone and transition from a glassy state to a viscous flow dynamic. Consequently, fusion occurs at the contact surface of the particles, promoting a strong bond, while also leading to the formation of small pores on the fracture surface.

To analyze the changes in the two components P1 and P2 in composite materials C during stepwise isothermal curing, the fracture surface of the samples, printed using P1, P2, and C, before and after heating was observed. Figure 10d–f displays the fracture morphologies of the samples after the first stage of heating. After heating at 85 °C for 1 h, in addition to small circular pores, there are also large circular pores with diameters in the range of tens of microns for the samples printed using P1. This indicates that the latent curing agent decomposed during heating, releasing a significant amount of gas, which expanded the pores as the epoxy resin melted, as shown in Figure 10d. In contrast, Figure 10e presents that the number of large pores in the samples printed using P2 was significantly less than that in the samples printed using P1, and some of the pores are irregular in shape. This may be attributed to the fact that no significant curing reaction occurred in the samples printed using P2, and most of the curing agents had not decomposed to generate gas. The fracture surface of the samples printing using C exhibited similar large pores to that of the samples printing using P1, as presented in Figure 10f, indicating that the P1 component within C had undergone a certain degree of the curing reaction.

Figure 10g–i shows the fracture morphologies of the samples after the second stage of heating. The fracture surface of the samples printed using P1 displayed more large pores compared with the first stage of heating. This could be attributed to the fact that the latent curing agent further decomposed at higher temperatures, producing more gas and forming more large pores. Similarly, the brittle fracture surface of the samples printed using P2 also exhibited more large pores after the second stage of heating. Additionally, the fracture surface of the samples printing using C displayed numerous large pores, including circular pores similar to those observed in the samples printing using P1 and irregular pores observed in the samples printing using P2, as presented in Figure 10i.

In addition, to investigate the effect of stepwise isothermal curing on the thermal properties, DSC tests were conducted on the samples printed using P1, P2, and C. As shown in Figure 11, the DSC curves of P1 and P2 before heating exhibited significant endothermic peaks at around 55 °C and 70 °C, indicating the existence of crystallization in the epoxy resin. In addition, significant exothermic peaks suggest that the samples did not complete the curing reaction during the CSAM process, indicating the need for post-heating curing treatment.

After heating at 85 °C for 1 h in the first stage, the endothermic peak of P1 disappeared and the exothermic peak significantly decreased, indicating that the curing reaction of P1 had been essentially completed in the first heating stage. In contrast, the endothermic peak of P2 disappeared, while the exothermic peak only showed slight changes, suggesting that the curing of P2 was not significantly activated in the first stage. After heating at 150 °C for 30 min in the second stage, the exothermic peak of P2 disappeared, indicating that the curing reaction of P2 was completed during the second stage of heating. From Figure 11, it is found that the thermal behavior of the samples printed using C exhibited an intermediate feature of the components, P1 and P2. Before heating, the peak temperature of C was approximately the average of the peak temperatures of P1 and P2. After the first heating, the exothermal peak of C decreased but did not completely disappear, indicating that the P2 component was not fully cured. After the second heating, the exothermic peak of C disappeared, implying that all the components had realized complete curing.

From the above analysis, it can be confirmed that the stepwise curing process induces sequential curing reactions of P1 and P2. During these curing reactions, gas is generated, leading to the formation of a porous structure.

### 3.3. Mechanical Property Evaluation

To investigate the influence of stepwise isothermal heating on the mechanical properties of the printed samples, TMA was conducted by using the proposed heating program illustrated in Figure 7. Figure 12 presents the changes in the indentation depth ratio of the samples printed using P1, P2, and C during the stepwise isothermal heating process, where the indentation depth ratio is defined as the ratio of the indentation depth to the thickness of the tested sample, to evaluate the sample deformation.

From Figure 12, it can be seen that the sample printed using P1 exhibited significant deformation in the first heating stage, and little deformation occurred in the second heating stage. In contrast, the sample printed using P2 displayed minimal deformation in the first heating stage but underwent severe softening deformation in the second heating stage. For the sample printed using C, although it experienced noticeable softening deformation in both stages, its final deformation degree was significantly lower than that of the samples printed using P1 and P2.

The unique behavior of the sample printed using C can be ascribed to the composite structure. During the first heating stage, the P1 component melted, while most of the P2 components remained solid, acting as rigid particles and providing reinforcement. This structural characteristic effectively reduced the deformation rate, resulting in a significantly smaller deformation compared with the sample printed using pure P1, as presented in Figure 12. Upon entering the second heating stage, the P1 component had already cured, and the P2 component began to melt. At this time, the cured P1 component played the role of reinforcing agent, leading to a smaller deformation at this stage compared with the sample printed using pure P2, as shown in Figure 12. In general, the excellent resistance to softening deformation exhibited by the sample printed using C throughout the entire heating process is due to the structural support provided by the unmelted P2 component and the cured P1 component in the first and second heating stages, respectively.

To assess the evolution of mechanical properties throughout the forming process, Vickers hardness tests were employed, as presented in Figure 13. Preliminary results showed no significant hardness differences between the samples printed using P1, P2, and C and their respective original powders. This observation suggests that although high-velocity particles impacted during the CSAM process leading to partial decomposition of latent curing agents, no substantial curing reaction was triggered.

During the subsequent stepwise heat treatment, the materials exhibited distinct behavioral patterns. After the first heating stage (85 °C, 1 h), P1 demonstrated a significant increase in Vickers hardness, indicating that the curing reaction effectively enhanced its mechanical properties. In contrast, P2 showed only a slight increase in hardness at this stage, suggesting that its primary curing reaction had not yet been initiated. In the second heating stage (150 °C, 30 min), the hardness of P1 increased marginally, indicating that its curing reaction was almost completed in the first stage. Conversely, P2 underwent a significant hardness enhancement during this stage, revealing the occurrence of its melting and curing processes. Notably, the hardness characteristics of the composite material C were gradually improved, correlated with the increase in the hardness of P1 and P2. This phenomenon reflects the gradual enhancement of mechanical properties in the composite material during the stepwise curing process. Ultimately, the hardness of composite material C after stepwise curing was comparable to that of samples cured using a low-rate heating process. It demonstrates that the stepwise curing technique not only obtains the same mechanical properties as the traditional low-rate heating methods but also significantly reduces the heating time.

## 4. Conclusions

This study investigated the feasibility of using thermosetting composite powders containing components with different curing temperatures for CSAM to avoid the softening and deformation issues encountered by thermosetting materials in additive manufacturing. Two types of single-component thermosetting powders, P1 and P2, as well as their composite powder, C, were utilized to prove the effectiveness of this approach. The main conclusions are presented as follows:

(1)The samples printed using P1, P2, and C remained solid throughout the CSAM process. During the progressive curing, the samples printed using C showed better deformation resistance than those using P1 and P2, and the mechanical properties after the progressive curing were comparable to those using low temperature-rate curing methods, while the curing time was reduced from 16.7 h to 1.5 h.(2)During the CSAM process, severe plastic deformation occurred because of the high-speed particle impact, accompanied by heat generation, which led to local melting to promote a good bond at the contact surface of the particles and form small pores.(3)During the stepwise curing process, the sequential curing reactions of P1 and P2 components in composite C allowed the uncured P2 and cured P1 to alternately remain solid, providing structural support and minimizing deformation in the stepwise curing process.

## Figures and Tables

**Figure 1 polymers-16-02416-f001:**
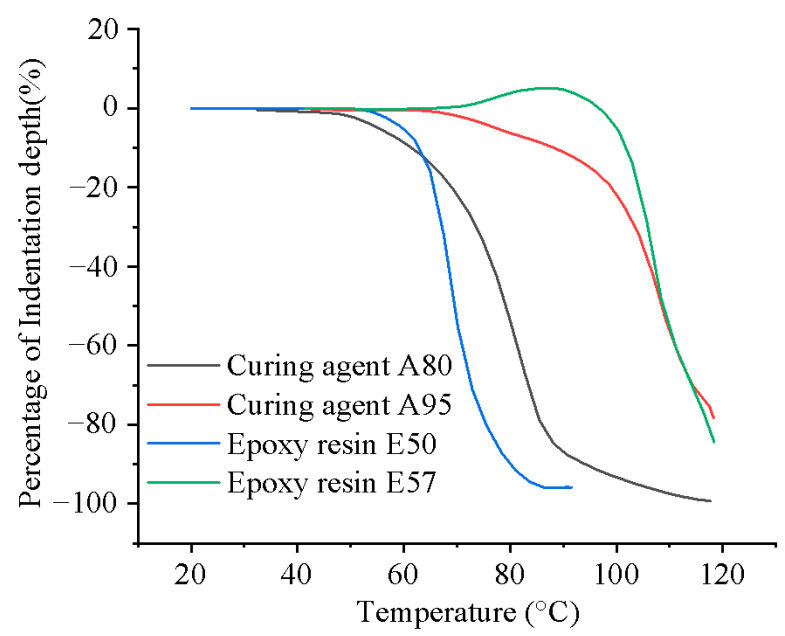
TMA of the as-received powder conducted at a constant heating rate of 10 °C/min.

**Figure 2 polymers-16-02416-f002:**
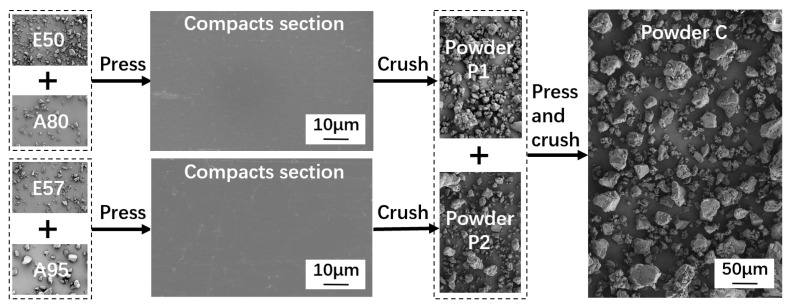
The powder fabrication process.

**Figure 3 polymers-16-02416-f003:**
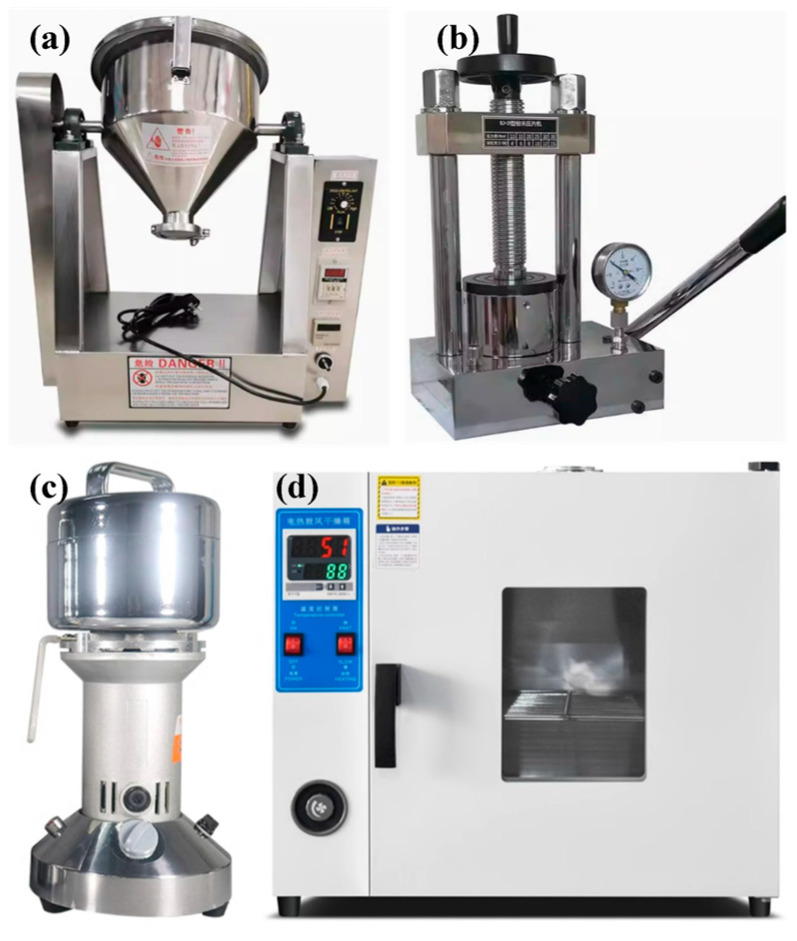
The equipment used in the powder fabrication process: (**a**) powder mixer, (**b**) hydraulic press, (**c**) blade grinder, and (**d**) forced air dryer.

**Figure 4 polymers-16-02416-f004:**
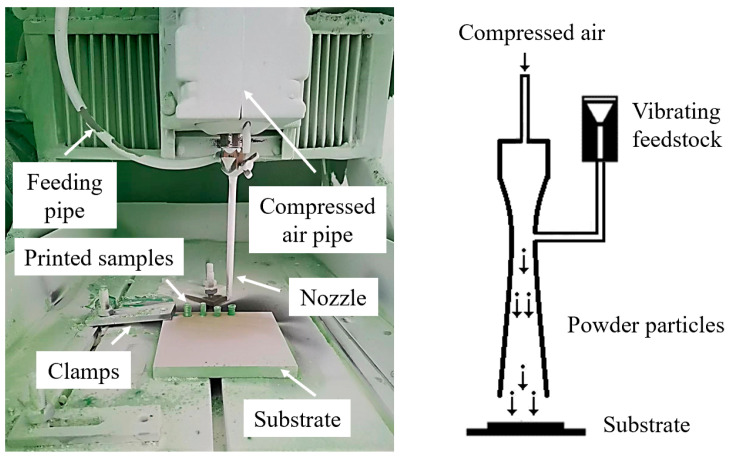
Cold spray system setup and schematic of CSAM process.

**Figure 5 polymers-16-02416-f005:**
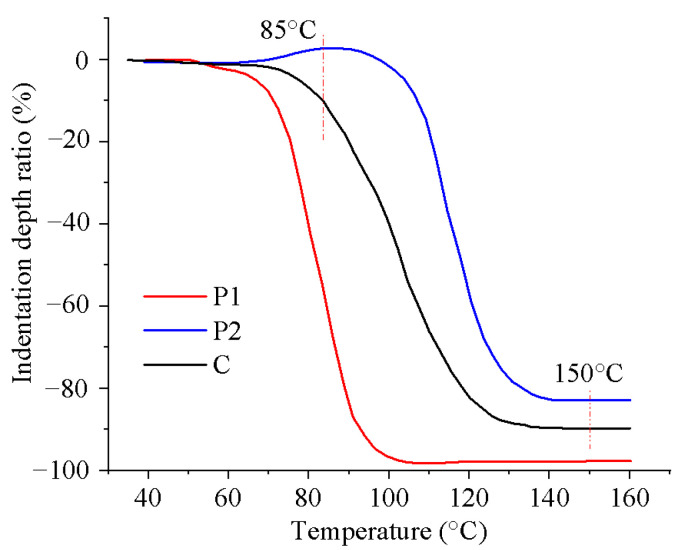
TMA of the prepared thermoset powder was conducted at a constant heating rate of 10 °C/min.

**Figure 6 polymers-16-02416-f006:**
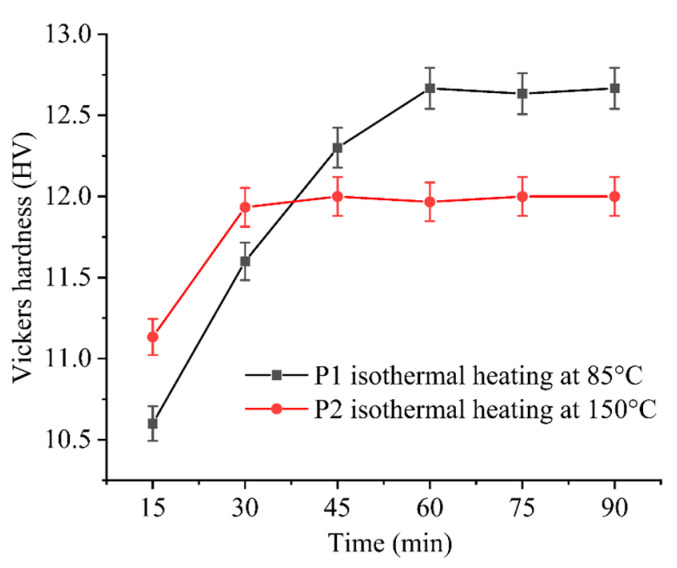
The evolution of Vickers hardness with heating time under isothermal heating for P1 and P2.

**Figure 7 polymers-16-02416-f007:**
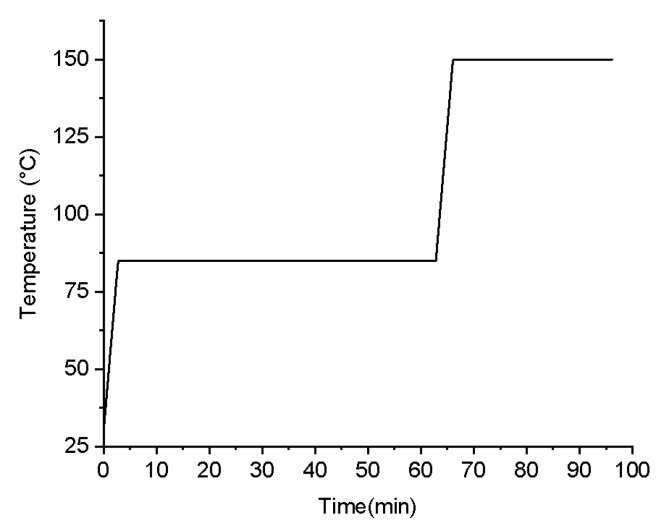
The stepwise isothermal heating process.

**Figure 8 polymers-16-02416-f008:**
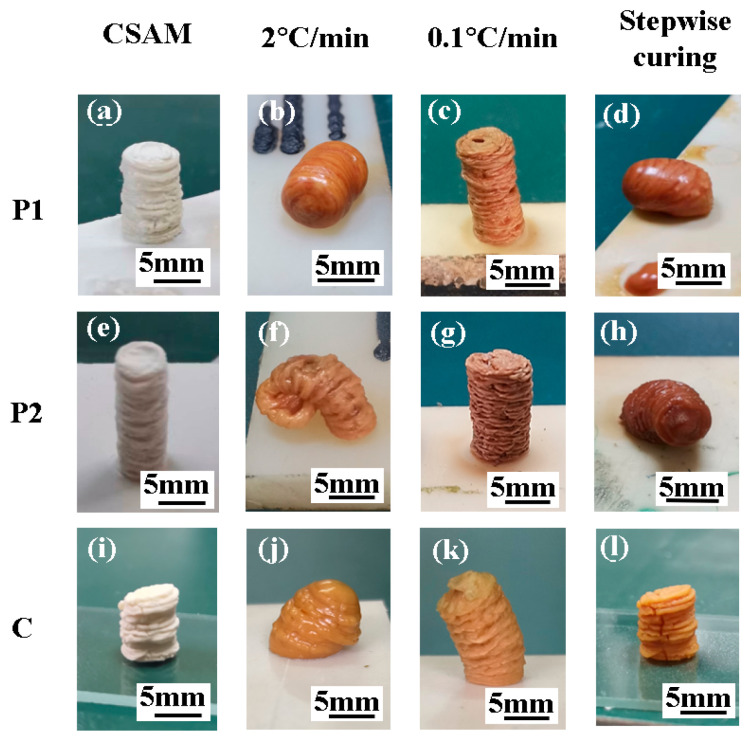
The morphology evolution of printed samples with different heating processes. (**a**–**d**) The morphology evolution of P1. (**e**–**h**) The morphology evolution of P2. (**i**–**l**) The morphology evolution of C.

**Figure 9 polymers-16-02416-f009:**
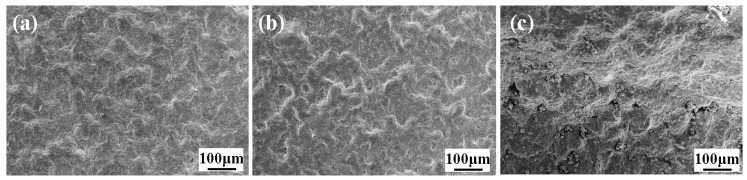
Surface morphologies of the samples printed using (**a**) P1, (**b**) P2, and (**c**) C.

**Figure 10 polymers-16-02416-f010:**
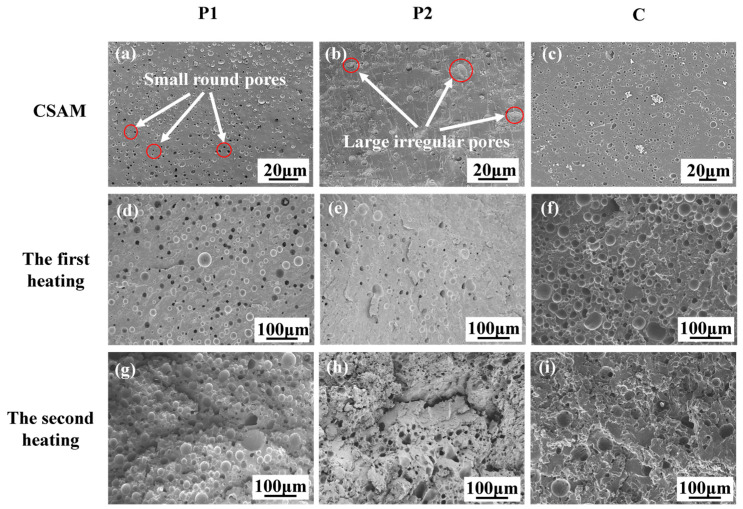
Fracture surface of the samples printed using P1, P2, and C: (**a**–**c**) before heating; (**d**–**f**) after the first heating at 85 °C for 1 h; and (**g**–**i**) after the second heating at 150 °C for 0.5 h.

**Figure 11 polymers-16-02416-f011:**
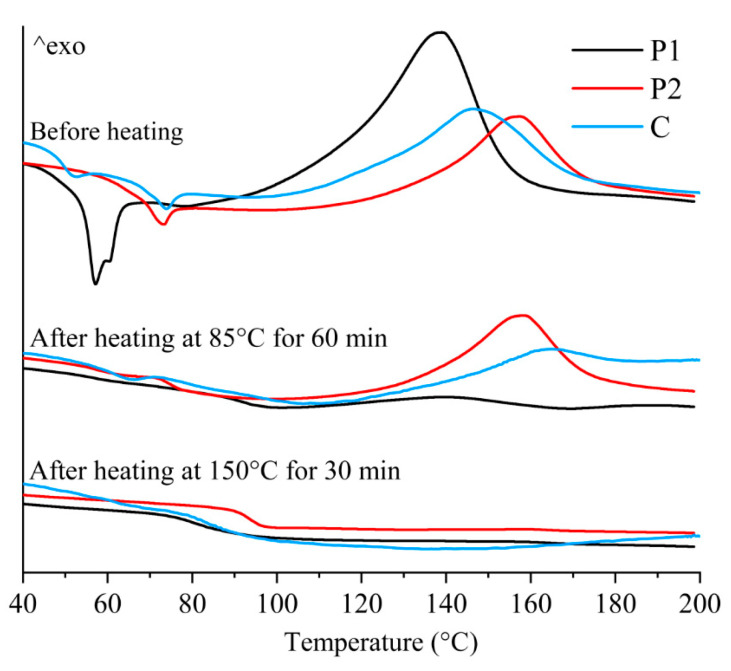
DSC curves of the samples printed using P1, P2, and C.

**Figure 12 polymers-16-02416-f012:**
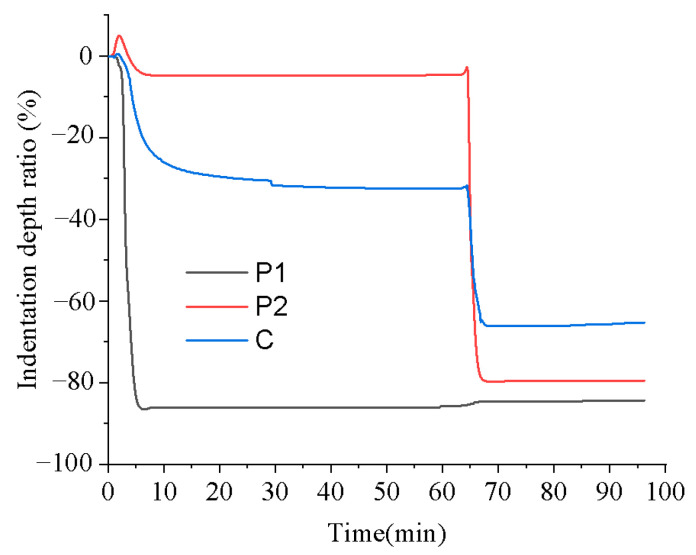
TMA of the printed samples with the stepwise isothermal heating process.

**Figure 13 polymers-16-02416-f013:**
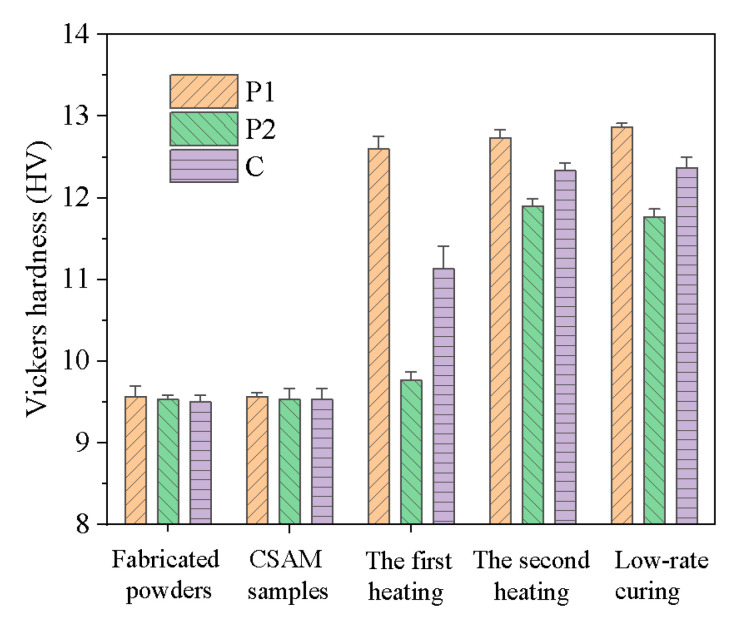
The Vickers hardness evolution of P1, P2, and C powders and printed parts before heating, after the first heating at 85 °C for 1 h, after the second heating at 150 °C for 0.5 h, and after full curing by slow heating from 50 °C to 150 °C with a rate of 0.1 °C/min.

## Data Availability

Data are contained within the article.

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
