# Peer review of "Solid State Additive Manufacturing of Thermoset Composites"

_polymers, 2024, doi:10.3390/polym16172416_

Round 1

Reviewer 1 Report

Comments and Suggestions for Authors

The synthesis of powders for additive manufacturing was done exceptionally well.

An example has been applied.

I have some comments and questions.

The conclusion must be rewritten. Conclusions should summarize the key findings from the paper, not explain theories or generate new content.

The abstract also needs revision. It is currently too generic and should provide a summary of the introduction, experimental methods, results, and discussion. This will help the audience understand the essence of your work and its significance from the abstract alone.

More details about the powder manufacturing processes are necessary. Include information about the machines used, the specific methods applied, the detailed conditions, and the mixing procedures. Additionally, clearly describe the differences between the three powders, P1, P2, and P3, including their thermal and mechanical properties. This will make the comparison more efficient and comprehensive.

Comments on the Quality of English Language

It is OK.

Author Response

The synthesis of powders for additive manufacturing was done exceptionally well.

An example has been applied.

I have some comments and questions.

The conclusion must be rewritten. Conclusions should summarize the key findings from the paper, not explain theories or generate new content.

The abstract also needs revision. It is currently too generic and should provide a summary of the introduction, experimental methods, results, and discussion. This will help the audience understand the essence of your work and its significance from the abstract alone.

More details about the powder manufacturing processes are necessary. Include information about the machines used, the specific methods applied, the detailed conditions, and the mixing procedures. Additionally, clearly describe the differences between the three powders, P1, P2, and P3, including their thermal and mechanical properties. This will make the comparison more efficient and comprehensive.

Authors’ Response: Thank you for the suggestions. We have carefully considered your comments and made the following modification to address your concerns:

  1. Conclusion Revision: We have rewritten the conclusion to summarize the key findings of this study.
  2. Abstract Revision: The abstract has been revised to provide a concise summary of the introduction, experimental methods, results, and discussion, highlighting the essence and significance of our work.
  3. Powder Manufacturing Process Details: We have included more detailed information about the powder manufacturing processes.
  4. Comparison of powders: A comparative analysis of the powders with respect to their thermal, microstructural, and mechanical properties is presented in the discussion section.

The corresponding modification is presented as follows:

Conclusion (on page 12)

This study investigated the feasibility of using thermosetting composite powders containing the components with different curing temperatures for CSAM to avoid the softening and deformation issues encountered by thermosetting materials in additive manufacturing, and two types of single-component thermosetting powders, P1 and P2, as well as their composite powder, C, were utilized to prove the effectiveness of this approach. The main conclusions are presented as follows:

(1) The samples printed using P1, P2 and C remained solid throughout the CSAM process. During the progressive curing, the samples printed using C showed better deformation resistance than those using P1 and P2, and the mechanical properties after the progressive curing were comparable to those using low temperature-rate curing methods, while the curing time was reduced from 16.7 h to 1.5 h.

(2) During the CSAM process, severe plastic deformation occurred because of the high-speed particle impact, accompanied by heat generation, which leaded to local melting to promote a good bond at the contact surface of the particles and form small pores.

(3) During the stepwise curing process, the sequential curing reactions of P1 and P2 components in composite C allowed the uncured P2 and cured P1 to alternately remain solid, providing structural support and minimizing deformation in the stepwise curing process.

Abstract (on page 1):

Softening and subsequent deformation are significant challenges in additive manufacturing of thermal-curable thermosets. This study proposes an approach to address these issues, involving the preparation of thermosetting composite powders with distinct curing temperatures, the utilization of cold spray additive manufacturing (CSAM) for sample fabrication, and the implementation of a stepwise curing for each component. To validate the feasibility of this approach, two single-component thermosetting powders P1 and P2 and their composite powder C were subjected to CSAM and stepwise curing. From the sample morphology observation and deposition/curing mechanism investigation based on thermomechanical analysis and differential scanning calorimetry, it is found that severe plastic deformation occurs during the CSAM process, accompanied by heat generation, leading to local melting to promote a good bond at the contact surface of the particles and form small pores. During the progressive curing, the samples printed using C demonstrate superior deformation resistance compared with those using P1 and P2, and the curing time is reduced from 16.7 h to 1.5 h, due to the sequential curing reactions of P1 and P2 components in composite C, allowing the uncured P2 and cured P1 to alternately remain solid for providing structural support and minimizing deformation.

Powder fabrication process (on page 3):

The powder preparation process was conducted in two distinct stages. Initially, single-component thermosetting powders P1 and P2 were fabricated using E50, E57, A80, and A95. Subsequently, composite powder C was fabricated by combining P1 and P2, as illustrated in Figure 2. The fabrication of P1 and P2 involved several steps: the epoxy resins and curing agents were mixed and then compressed, followed by pulverization in water and a drying process. The preparation of composite powder C followed a similar procedure, with P1 and P2 first mixed at a 1:1 volumetric ratio, then compressed, pulverized, and dried. All operations were conducted at room temperature to prevent curing reactions.

For the mixing process, a small drum-type powder mixer equipped with stirring blades was employed. The device operated with a bi-directional rotation frequency of 0.5 Hz for a duration of 1 h. Compression was performed using a manual hydraulic press with a cylindrical mold of 50 mm internal diameter. The mixed powder was poured into the mold and subjected to a pressure of 30 MPa for 2 min, resulting in cylindrical bulks. As observed in the cross-section of the compacted powder in Figure 2, the powder particles were fully deformed and interlocked. Pulverization was carried out using a 600 W blade grinder equipped with cross-shaped rotating blades. The cylindrical bulks were first broken into pieces with a hammer, and then processed in the grinder along with water. The grinder was activated three times for 10 s each time, with each grinding interval of 1 min. Water was added to prevent curing reactions due to a rapid increase in temperature during grinding. After pulverization, water was first filtered out of the ground material with a filter cloth. The resulting wet powder was then placed in a forced air dryer at 30 °C for 24 h to ensure that the water was completely removed. The average particle sizes of P1, P2 and C powders were between 45 and 50 μm and Figure 2 depicts their irregular morphologies.

Reviewer 2 Report

Comments and Suggestions for Authors

The manuscript investigated the solid-state additive manufacturing of thermoset composites. It is well-written. I suggest it be accepted after some minor revisions. The comments are as follows.

1.        The authors reported that all operations were conducted at room temperature. However, they used water as the grinding solvent and water was subject to a drying process. It should be explained.

2.        The figure caption of Fig.9 should describe which belong to P1 and P2 samples.

3.        The microstructures of C samples should be provided in Fig.9.

4.        It will be better to provide DSC curve of sample C in Fig.10.

Author Response

The manuscript investigated the solid-state additive manufacturing of thermoset composites. It is well-written. I suggest it be accepted after some minor revisions. The comments are as follows.

  1. The authors reported that all operations were conducted at room temperature. However, they used water as the grinding solvent and water was subject to a drying process. It should be explained.

Authors’ Response: Thank you for the reminding. In the revised manuscript, we have added a detailed explanation to clarify that “Water was added to prevent curing reactions due to a rapid increase in temperature during grinding.” Regarding the drying process, we have provided more information “After pulverization, water was first filtered out of the ground material with a filter cloth. The resulting wet powder was then placed in a forced air dryer at 30 °C for 24 h to ensure that the water was completely removed.”

  1. The figure caption of Fig.9 should describe which belong to P1 and P2 samples.

Authors’ Response: Thank you for the reminding. In the revision, we have labeled P1, P2, and C in Figure 9.

  1. The microstructures of C samples should be provided in Fig.9.

Authors’ Response: Thank you for the comment. We have added the microstructures of C samples in Fig.9 and the corresponding morphology analysis on page 9: “The fracture surface of the samples printing using C exhibited similar large pores to that of the samples printing using P1, as presented in Figure 9(f), indicating that the P1 component within C had undergone a certain of curing reaction.” and “Additionally, the fracture surface of the samples printing using C displayed numerous large pores, including circular pores similar to those observed in the samples printing using P1 and irregular pores observed in the samples printing using P2, as presented in Figure 9(i).” Moreover, we revised the numbering of the images in Figure 9.

  1. It will be better to provide DSC curve of sample C in Fig.10.

Authors’ Response: Thank you for the suggestion. We have revised the manuscript to include the DSC curve of the samples printed using C in Figure 10 and added the corresponding analysis on page 10: “From Figure 10, it is found that the thermal behavior of the samples printed using C exhibited an intermediate feature of the components, P1 and P2. Before heating, the peak temperature of C was approximately the average of the peak temperatures of P1 and P2. After the first heating, the exothermal peak of C decreased, but did not completely disappear, indicating that the P2 component was not fully cured. After the second heating, the exothermic peak of C disappeared, implying that all the components had realized complete curing.”

Reviewer 3 Report

Comments and Suggestions for Authors

In this paper, the authors tried to 3D print thermosetting materials using the cold spray technique. The subject is interesting, and the experiments and results are presented in an acceptable manner. However, the results of this paper can be considered as proof of concept, and further investigation is required for applicability in academic research or industries. Nonetheless, this paper can be suitable for publication after the authors address the following questions:

1.      Some of the abbreviations are defined multiple times. Please revise the text. For example, CSAM is defined at lines 15 and 51.

2.      As mentioned, the nozzle only moved vertically during the printing process. The speed of this movement is an important parameter and will affect the printing quality and the mechanical properties. Could you provide some explanation regarding this? How did you choose this parameter?

3.      The speed of the particles when they leave the nozzle directly affects the print quality and the mechanical properties. This parameter can be controlled by changing the applied air pressure. However, the authors only considered one pressure and did not examine the effects of this parameter on the print quality and mechanical properties. Could you elaborate more on how you chose this parameter?

4.      At the end of Section 3.1, the authors claimed that using powder C and stepwise curing led to acceptable results. Considering that the printed structure and printing path is very simple, can we conclude the same for more complex geometries as well?

5.      Is it possible to use this method with smaller nozzles? What would be the challenges and achievements?

6.      Considering the cylindrical shape of the samples, it was possible to do the compression test to examine the samples' compressive properties. Hardness is not a suitable parameter to measure the mechanical properties of the printed samples. I suggest the authors try to perform the compression test on the printed samples as well.

Author Response

In this paper, the authors tried to 3D print thermosetting materials using the cold spray technique. The subject is interesting, and the experiments and results are presented in an acceptable manner. However, the results of this paper can be considered as proof of concept, and further investigation is required for applicability in academic research or industries. Nonetheless, this paper can be suitable for publication after the authors address the following questions:

  1. Some of the abbreviations are defined multiple times. Please revise the text. For example, CSAM is defined at lines 15 and 51.

Authors’ Response: Thank you for pointing it out. In the revision, we have removed the multiple definition of CSAM.

  1. As mentioned, the nozzle only moved vertically during the printing process. The speed of this movement is an important parameter and will affect the printing quality and the mechanical properties. Could you provide some explanation regarding this? How did you choose this parameter?

Authors’ Response: Indeed, the speed of this nozzle movement is an important parameter and will affect the printing quality and the mechanical properties. For the experimental setup, it is found that the powder accumulation rate during deposition is approximately 2 mm/s when the air pressure is 0.8 MPa. To maintain a consistent stand-off distance of 10 mm between the nozzle exit and the top surface of the deposited sample, we set the vertical movement speed of the nozzle to match this accumulation rate.

To provide a comprehensive description of the nozzle movement, we revised the last paragraph of Section 2.3 on page 4 as follows: “Cylindrical samples for P1, P2, and C were fabricated during the CSAM process at room temperature. The nozzle was initially fixed at a specific point on the substrate for cold spraying. It is noticed that during the powder deposition process, the accumulation rate of the deposited material was approximately 2 mm/s when the air pressure was 0.8 MPa. To maintain a constant stand-off distance between the nozzle exit and the top surface of the samples being deposited (10 mm in this study), the nozzle movement was controlled vertically with the speed of 2 mm/s, facilitating the layer-by-layer construction of the cylindrical samples.”

  1. The speed of the particles when they leave the nozzle directly affects the print quality and the mechanical properties. This parameter can be controlled by changing the applied air pressure. However, the authors only considered one pressure and did not examine the effects of this parameter on the print quality and mechanical properties. Could you elaborate more on how you chose this parameter?

Authors’ Response: We thank the reviewer for pointing it out. Indeed, the print quality and the mechanical properties will be affected by the selection of the printing parameters of the CSAM system, such as the applied air pressure, the nozzle movement speed and printing temperatures, etc. In this study, the primary objective is to investigate the feasibility of using thermosetting composite powders containing the components with different curing temperatures for CSAM to avoid the softening and deformation issues encountered by thermosetting materials in additive manufacturing. We just adopted one combination of the printing parameters, i.e., the applied air pressure is 0.8 MPa, which is maximum air pressure of the customized CSAM system to achieve sufficient powder deposition, the nozzle movement speed is 2 mm/s to match the accumulation rate, and the samples were printed at room temperature.

Thank you for the reminding. For the future work, we will investigate the influence of the printing parameters on the print quality and the mechanical properties for the CSAM of thermosetting composite powders, laying the foundation for their potential application.

  1. At the end of Section 3.1, the authors claimed that using powder C and stepwise curing led to acceptable results. Considering that the printed structure and printing path is very simple, can we conclude the same for more complex geometries as well?

Authors’ Response: Thank you for the comment. In this study, only cylindrical samples were utilized to validate the feasibility of using thermosetting composite powders with different curing temperature components for CSAM to avoid the softening and deformation problems. It is found that the samples printed using thermosetting composite powders show superior deformation resistance, and the curing time can be reduced significantly, confirming the effectiveness of the proposed approach.

Actually, the printing structure does not affect the above conclusions, while the printing parameters will influence the print quality and the mechanical properties. Therefore, our future work will focus on exploring the optimal combination of the printing parameters.

  1. Is it possible to use this method with smaller nozzles? What would be the challenges and achievements?

Authors Response: Thank you for your suggestion. In this study, we just adopted the nozzle with a diameter of 5 mm. It is noticed that smaller nozzle size can enhance the precision and resolution of the printed results. However, implementing smaller nozzles in CSAM will cause several challenges, such as powder flow issues, and low deposition efficiency. The main purpose of this study is verifying the feasibility of using thermosetting composite powders with different curing temperature components for CSAM, and the nozzle size effect is not considered.

  1. Considering the cylindrical shape of the samples, it was possible to do the compression test to examine the samples' compressive properties. Hardness is not a suitable parameter to measure the mechanical properties of the printed samples. I suggest the authors try to perform the compression test on the printed samples as well.

Authors’ Response: Thank you for the reminding. In this study, the sample preparation for compression testing is challenging due to the wrinkled surface and brittleness of the printed samples. Vickers microhardness can not only reflect the curing degree during the CSAM and stepwise curing process, but also reflect the mechanical properties of the printed samples in some respects. Hence, Vickers microhardness is determined in this study.

With the improvement of the print quality by optimizing the combination of the printing parameters, compression tests will be considered in our future studies, for exploring the potential applications of the proposed approach.

Round 2

Reviewer 1 Report

Comments and Suggestions for Authors

Thank you for your effort in improving the manuscript.

I have a few additional questions:

  1. Please ensure that P1, P2, and C are explained solely in the text for all figures.

  2. All names, models, and countries of origin for the equipment, devices, and instruments used to analyze data and conduct experiments must be mentioned.

  3. The detailed conditions and parameters for each experiment must be clearly documented.

  4. Include a figure of the equipment used to synthesize the powder, along with a thorough description of this equipment.

Additionally:

  • What type of machine did you use to analyze the microstructures and fractures?
  • What are the samples, and how did you analyze the fractures? How were the samples prepared, and which standards were followed?

Author Response

  1. Please ensure that P1, P2, and C are explained solely in the text for all figures.

Authors’ Response: Thank you for the reminding. In Section 3.2, we added more details to the deposition and curing mechanism analysis. For Figure 9, we added “It can be seen that all the samples printing using P1, P2, and C had a number of sharp ridges and valleys on the deposited surface.” and for Figure 10, we added “A number of pores were found throughout the fracture surface of the samples, where the samples printing using P1 had the highest pore density, and the ones printing using P2 had the lowest pore density.”

  1. All names, models, and countries of origin for the equipment, devices, and instruments used to analyze data and conduct experiments must be mentioned.

Authors’ Response: Thank you for the reminding. In the revision, we have added the description of names, models, and countries of all the equipment utilized in this study, and marked them out in yellow.

  1. The detailed conditions and parameters for each experiment must be clearly documented.

Authors’ Response: Thank you for the reminding. In the revision, we had clarified the conditions and parameters for each experiment, and the modifications are highlighted in yellow.

  1. Include a figure of the equipment used to synthesize the powder, along with a thorough description of this equipment.

Authors’ Response: Thank you for the comment. In the revision, we have added a figure (i.e., Figure 3) to present the devices utilized for the powder preparation, and highlighted their names, models, and countries of origin.

Additionally:

  • What type of machine did you use to analyze the microstructures and fractures?

Authors’ Response: Thank you for the reminding. We have added the machine description in the “Microstructural analysis” section on page 6: “The surface features and fracture surface morphologies of the thermosetting powders and cold-sprayed samples before and after heating were characterized using the Hitachi SU-1510 (Japan) scanning electron microscope (SEM).”

  • What are the samples, and how did you analyze the fractures? How were the samples prepared, and which standards were followed?

Authors’ Response: Thank you for the reminding. We have added the sample preparation and analysis method in the “Microstructural analysis” section on page 6: “Note that all the samples were fabricated using CSAM with powders P1, P2, or C, and the corresponding fracture surface was obtained by breaking the samples using pliers. After coating a thin layer of platinum on the sample surface to enhance conductivity, the SEM observation was conducted following the ISO 16700:2016 standard.”
